# A Variable Stiffness Electroadhesive Gripper Based on Low Melting Point Alloys

**DOI:** 10.3390/polym14214469

**Published:** 2022-10-22

**Authors:** Chaoqun Xiang, Wenyi Li, Yisheng Guan

**Affiliations:** 1Jiangsu Provincial Key Laboratory of Special Robot Technology, Hohai University, Changzhou Campus, Changzhou 213022, China; 2School of Electromechanical Engineering, Guangdong University of Technology, Guangzhou 510006, China

**Keywords:** soft gripper, variable stiffness, electroadhesion, soft electroadhesive, soft pneumatic gripper

## Abstract

Electroadhesive grippers can be used to pick up a wide range of materials, and those with variable stiffness functionality can increase load capacity and strength. This paper proposes an electroadhesive gripper (VSEAF) with variable stiffness function and a simple construction based on low melting point alloys (LMPAs) with active form adaptation through pneumatic driving. Resistance wires provide active changing stiffness. For a case study, a three-fingered gripper was designed with three electroadhesive fingers of varied stiffness. It is envisaged that these electroadhesive grippers with variable stiffness would extend the preparation process and boost the use of electroadhesion in soft robot applications.

## 1. Introduction

To operate safely in unstructured environments and in collaboration with humans, a new generation of robots must be created; therefore, soft robots comprised of soft materials invented recently have a high degree of deformation, and their interactions with humans and objects are more flexible and compliant than those of traditional stiff robots [1,2,3]. In order to increase production efficiency, the agriculture, food, and medical industries must automate a number of complex grasping tasks involving fragile materials (such as grabbing fruits, vegetables, and biological tissue) [4,5]. Traditional robot end effectors, on the other hand, are generally made of rigid materials that make it difficult to interface with objects and have limited adaptability to curved surfaces, making it difficult to carry out such operational tasks [6,7].

Electroadhesion (EA) is a promising adhesion mechanism and material handling technique that is typically powered by a high voltage (commonly in the range of 0.2 to 20 kilovolts) between pairs of electrodes [8,9,10,11]. The key benefits are (1) the comparatively low energy consumption of EA systems and (2) the simplicity with which EA grippers can be made due to their fundamental architecture.

EA requires close contact with an object in order to generate sufficient gripping force. EA with shape adaptation characteristics can increase gripping efficiency. Guo et al. proposed a morphologically adaptive EA end effector, the PneuEA gripper [11], which was able to grab both flat and convex items. Due to air leakage, this type of gripper is unable to maintain appropriate contact conditions and may lose efficiency. There is currently no EA end effector that supports both active deformation and acceptable contact conditions.

To address the lack of rigidity and low load capacity of soft robots, researchers proposed the notion of variable stiffness. Soft robots equipped with variable stiffness functionality can boost safety when considering physical human–robot cooperation [12,13] and load capacity [14]. Variable stiffness methods include the blocking principle [15,16], the principle of variable stiffness structures, and the principle of smart materials. The blocking principle is based on negative pressure [17]; through air suction, materials within a soft and enclosed chamber are compressed, and their stiffness is increased, which ordinarily requires a large powder air pump and air system and increases the system’s complexity. Variable stiffness structures are most commonly found in rigid robots; however, variable stiffness robots, such as the variable stiffness continuum arm [18], have emerged in abundance in recent years. Soft robots with variable stiffness structures typically have a large bulk, which limits the downsizing of application situations. The smart materials principle employs smart functionality materials, such as shape memory polymers [19,20] and shape memory alloys [21], and these materials loaded with memory effect may modify stiffness in response to a given amount of stimuli; however, this method will introduce control challenges. Using phase-change material is a novel variable stiffness technique, and robots based on this material can alter their stiffness by joule heating directly, resulting in simple construction and straightforward application [22]. In this paper, low-melting-point alloys with good conductivity and variable stiffness characteristics [22] are used as the variable stiffness materials.

Electrostatic adhesion (EA) is generated by applying a high voltage (typically between 0.2 and 20 kilovolts) across pairs of electrodes embedded in a dielectric material. This high electric field produces an electrically controllable adhesion force on conductive, semi-conductive, and insulating substrates [23]. EA with an active form adaptive functionality can extend an object’s contact area, resulting in better gripping effectiveness. Variable stiffness EA has been researched [23], which may alter its stiffness based on the gripping task. However, it cannot actively alter its deformation or deform to complex shapes in order to grab items with complex surfaces. Consequently, there is still no simple, viable, and convenient method for designing an EA with variable stiffness.

To address the soft robots’ lack of rigidity and limited load capacity, this paper proposes an electroadhesive gripper with variable stiffness and simple construction (VSEAF) based on low melting point alloys (LMPAs) with active form adaptation via a pneumatic drive. A soft pneumatic gripper of electrostatic adsorption with a variable stiffness function was proposed in this work to meet the requirement of good adaptability between the electro-adhesive gripper and substrate. The contents are organized as follows: the specifics of the VSEAG’s design and manufacture are discussed in Section 2; in Section 3, the variable stiffness test, electroadhesive force test, and gripping task case study are described; in Section 4, conclusions and future study are described.

## 2. Materials and Methods

### 2.1. VSEAF Design

We intend to develop a variable stiffness electroadhesive finger (VSEAF) that can adjust its stiffness during pneumatic actuation in order to conform to different shapes. The soft pneumatic finger is constructed with an inflated chamber, low melting point alloys (LMPAs, indium tin alloy, melting point 47 °C), and electric resistance wires (4 Ω/m, Creative, Shen zhen, China) for this purpose (see Figure 1). In order to produce bending, an inflated chamber is utilized.

The VSEAF exhibits a rigid state when low melting point alloys are solid and a soft state when low melting point alloys are liquid. Low-melting-point alloys can be used to generate EA force in order to boost the VSEAF’s grasping capacity, and the electric resistance wire can be used to heat low temperatures. As a result, VSEAF can not only grip objects while pneumatically actuated, but it can also generate electroadhesive force to grasp objects of different shapes and sizes. The article’s gripper is pneumatic, and its mobility is controlled by a pneumatic system. This gripper provides a smooth action due to its usage of a pneumatic triplet.

The VSEAF workflow is as follows: VSEAF approaches the object to be grasped, the resistance wire is heated by electricity, the gripper enters a state of low stiffness, and the VSEAF chamber is inflated and bent to conform to the object to be grasped. The power to the electric resistance wire is then turned off, and when the low melting point alloy has cooled and solidified, the VSEAF will be in a rigid state. Lastly, a high voltage is given to the low melting point alloy to initiate electrostatic adsorption, allowing the gripper to grasp and hoist the clamped object with stability and dependability.

### 2.2. VSEAF Fabrication

The low-melting-point alloys are used to create a soft pneumatic gripper and a variable stiffness EA pad for the VSEAF device. The proposed method for manufacturing the VSEAF grippers is simple, as shown in Figure 2. The fabrication process consists of these three fundamental steps: Figure 2a–c illustrates how to make the variable stiffness electroadhesive pad. Figure 2d illustrates how to mould the soft pneumatic gripper, and the final step is to combine the two. The three steps are as follows:

**Moulding the variable stiffness EA pad.**Figure 2a demonstrates the construction and utilization of three 3D-printed parts (two comb-shaped components and one 5-mm-deep mold). First, the silicone Ecoflex 30 mixture was degassed and poured into the completed molds. Second, the two comb-shaped parts were extracted from the cured silicone molds, creating a groove for the low melting point alloys. This groove is utilized in the manufacturing of EA electrodes. Figure 2c depicts the geometry of the electrodes employed in this work, which have an effective electrode area of 111 mm by 20 mm. By heating to temperatures over the melting point (47 °C), pouring into comb-shaped grooves, and then cooling in cold water, an EA pad made of a low melting point and high conductivity was created. Thirdly, as shown in Figure 2b, the resistance wire is wrapped back and forth on the comb groove to form an S-shaped appearance. The heat generated by the solution is sufficient to rapidly melt low-melting-point alloys. After the dried comb-shaped LMPAs were positioned for EA pad sealing, a silicone Ecoflex 30 slurry was added to the groove. The EA pad with variable stiffness is shown in Figure 2c.

**Molding a pneumatic soft gripper.** Using a 3D printer (Raise3D, Raised E2, Shanghai Fusion Tech Co.,Ltd., Shanghai, China) and PLA filament with a 3 mm diameter, two moulds were created in three dimensions. Before filling these two moulds, equal amounts of silicone Ecoflex 30 (Smooth-On Inc., Pennsylvania, USA) were blended and weighed. The Ecoflex mixture was then degassed for twenty minutes. The mould was then cured in a 50 °C oven for two hours.

**Integration of the EA pad with the soft pneumatic gripper**. The soft pneumatic gripper was affixed to the variable stiffness EA pad using a silicone Ecoflex 30 combination, and the assembly was cured in a 50 °C oven for two hours. Figure 1b depicts the combination of the soft pneumatic actuator and the variable stiffness EA pad.

### 2.3. Geometric Parameter Selection and Repeatability Test EA Pad Fabrication

#### 2.3.1. Repeatability Test for EA Pad Fabrication

We developed tangential EA force measurement equipment, as seen in Figure 3, to validate the repeatability of the proposed variable stiffness EA manufacturing method. Papers were utilised as the substrate for the tangential EA force test, and after being charged for 30 s at different voltages using a high-voltage amplifier source (HVA, EMCO E60, XP Power Ltd., New Hampshire, USA), they were absorbed on the EA pad. The output of the HVA was controlled by a driver, an Arduino, and a PC. The weight of the paper was described as the tangential EA force generated by VSEAF. For low-melting-point alloys, a variable stiffness EA pad measuring 121 mm in length, 30 mm in width, and 3.5 mm in height was used. We also created an EA electrode spacing with 2 mm, 3 mm, and 4 mm, respectively. On these three EA pads, 3.2 kV, 3.8 kV, and 4.4 kV were applied, and Figure 4 illustrates the tangential EA force of VSEAF.

For the fabrication repeatability test, EA pads with three unique spacings were fabricated three times; notably, EA pads with three unique spacings were created three times for the fabrication repeatability test, resulting in the manufacturing of nine EA pads. Three tests were conducted for each EA pad, and these experiments were conducted in a clean, constrained setting. In addition, tests were done at a temperature of 22.6 °C ± 0.1 °C, relative humidity of 55% ± 1%, and ambient pressure of 1025.5 hPa ± 0.2 hPa using a weather station. There was a maximum relative difference of 4.5% in the average tangential EA forces collected from the nine EA pads, as shown in Figure 4b. EA is one kind of parallel capacitor, and the space between two electrodes, s, is constant. The spacing, s, between three EA pads was chosen to be 2 mm, 3 mm, and 4 mm, respectively. The greatest tangential EA forces for 2 mm, 3 mm, and 4 mm electrode spacing on the EA pads are 0.43 N, 0.48 N, and 0.512 N, respectively. The electrostatic tangential EA forces difference for three EA pads with identical specifications is between 0.05 N, indicating that the EA pad manufacturing process is very reliable and can manage error within a suitable range. Using cutting-edge EA pad fabrication methods, such as flexible printed circuit board manufacturing processes, might minimize this discrepancy even more [24].

#### 2.3.2. EA Pad Geometric Parameter Selection

As illustrated in the insets of Figure 5a,b, the geometric selection of the EA pair unit was based on an experimental investigation employing a customised EA electrode design, manufacturing, and EA force test platform. Here, empirical EA electrode geometry selection was conducted. The EA pad was positioned vertically for the tangential EA force test and horizontally and facing the ground for the normal EA force test using the same force measuring device. Again, papers were used as the test substrate, and after 30 s of charging at different voltages, the papers were absorbed by the EA pad.

The test results for tangential and normal EA forces are shown in Figure 5a,b, respectively. Given equal conditions, it is evident that the tangential electrostatic EA forces, in particular, diminish considerably as electrode spacing increases. When electrode spacing increases, fewer electrodes are present in the adsorption zone, which decreases the EA force.

As electrode spacing rises, so does the breakdown voltage of the EA pad; for example, electrode spacings of 2 mm, 3 mm, and 4 mm corresponded to breakdown voltages of 3.2 kV, 3.8 kV, and 4.4 kV, respectively. Clearly, in a certain range, an EA pad may produce a larger EA force with a wider electrode spacing, but to achieve the same EA force, an EA with a wide electrode spacing requires a higher excitation voltage. Similar to how stronger EA forces need more costly high voltage amplification equipment and higher prices, a higher voltage also incurs more safety risks; therefore, increasing the electrode spacing is not the sole approach to enhance adsorption capacity. The maximum tangential force generated by EA pads with 2 mm, 3 mm, and 4 mm electrode spacing is 0.45 N, 0.46 N, and 0.48 N, respectively. Furthermore, 0.07 N, 0.09 N, and 0.10 N are the maximum normal forces produced by EA pads with 2 mm, 3 mm, and 4 mm electrode spacing, respectively. As a result, the 4 mm electrode spacing was chosen for the EA pad design in order to maximize safety and EA force and provide the best results.

Tangential EA forces of EA pads with various dielectric layer thicknesses were investigated since the thickness of the dielectric layer can impact the output of the EA force. Four EA pads with dielectric layer thicknesses of 0.4 mm, 0.6 mm, 0.8 mm and 1 mm were constructed accordingly. Five tangential force tests were done for each EA pad using the tangential force measuring instrument. It can be seen in Figure 5c that the EA force has an obvious downward trend with the increase in the thickness of the dielectric layer, especially when the insulation layer thickness is 1 mm. EA pads with dielectric layer thicknesses of 0.4 mm, 0.6 mm, 0.8 mm and 1 mm are driven at the same voltage of 4.4 kV, and the resultant tangential EA forces for the four kinds of EA pads were 0.55 N, 0.489 N, 0.41 N and 0.27 N, respectively. Compared to EA pads with a 1 mm dielectric layer, EA pads with a 0.4 mm dielectric layer may produce 49% more tangential EA force. Considering the transition of a low melting point alloy from liquid to solid, the heat bilge cold shrink phenomenon exists; therefore, we must choose a certain amount of security when determining the thickness of the insulating layer in the event that the insulating layer is compromised due to the low melting point alloy’s large expansion volume when it is liquid. Consequently, 0.6 mm was chosen for the dielectric layer.

## 3. Results

### 3.1. Variable Stiffness Test

The VSEAF was placed horizontally on a PLA frame, as illustrated in Figure 6, to demonstrate the gripper’s variable stiffness function. Figure 6 demonstrates that the VSEAF is stiff when the temperature of LMPAs is below their melting point (47 °C). Figure 6b depicts how the VSEAF softened due to the fluid nature of the LMPAs at temperatures above their melting points. As demonstrated in Figure 6c [25], the VSEAF reacted similarly to a typical pneumatic gripper after being inflated to a soft state at 40 kPa.

The VSEAF stiffness predominantly occurs in two states: the rigid state for low melting point alloys in solid form and the soft state for low melting point alloys in liquid form. In order to evaluate the variable stiffness characteristics, a stiffness performance test rig was set up, the bending angles were captured using a camera, and a coordinate paper was put on a specific stiffness performance test apparatus, as seen in Figure 7. The VSEAF was horizontally positioned at a frame. The low melting alloys were heated by an Arduino MEGA 2560 connected to a PC via an electric resistance wire. By applying 11 V voltage to both ends of the resistance wire for one minute, it is possible to totally melt the low-melting-point alloy. The alloy with a low melting point may resolidify at normal temperature. The VSEAF will be subjected to a pressure of 30 kPa prior to the measurement. A load was tied to the end of the VSEAF, and the load was raised from 0 to 0.49 N in increments of 0.098 N using the displacement-force curves of solid and liquid states in the *x*-axis of the low-melting-point alloy (Figure 7c) and the displacement-force curves of solid and liquid states in the *y*-axis of the low-melting-point alloy (Figure 7d). Thus, two sets of stiffness data can be produced.

In this experiment, stiffness is represented by the displacement brought about by one unit of force, which is stiffness’s reciprocal (flexibility). The reciprocal of stiffness and flexibility shows that the stiffness and flexibility have an inverse relationship with the slope of the curve. Figure 7c,d demonstrate that the rigid state of the low melting alloy exhibits much greater VSEAF stiffness than the liquid state in both the X and Y axes. In the x direction, VSEAF has a stiffness of 0.143 N/mm in the rigid state and 0.016 N/mm in the soft state. In the y direction, the VSEAF is 0.207 N/mm stiff, and the soft state is 0.007 N/mm soft. In conclusion, the design of the VSEAF can adjust its stiffness in a certain range.

Studying the bending performance of the VSEAF is very significant since the bending performance of soft fingers is a key indicator for revealing the workspace of a soft pneumatic gripper. Figure 8 illustrates how the pressure within the VSEAF was adjusted from 0 kPa to 60 kPa in increments of 10 kPa, while the bending performance was vertically set at a frame. Three tests were run for each inflating pressure of the VSEAF.

The measurement and comparison of the bending angle corresponding to different inflation pressures are shown in Figure 8b. As shown, the low melting point alloy exhibits a substantially larger shift in bending angle while it is liquid than when it is solid. However, the change in bending angle makes it easier to detect the considerable influence of the low melting point alloy’s solid-liquid state conversion on the VSEAF bending performance. It is obvious that the bending angle change of VSEAF during the liquid state of the low melting point alloy is substantially greater than that during its solid state. The important effect of the low melting alloy’s solid-liquid state conversion on the bending performance of VSEAF, however, may be more effectively shown by the change in bending angle in the interim. The VSEAF in its soft form demonstrates bending movement due to gravity.

### 3.2. Demonstration of Gripping Capability

#### 3.2.1. Demonstration of EA Gripping Capability

The EA function allows the VSEAF to safely grasp both convex and flat objects, unlike other pneumatic grippers. The ability of the VSEAF to grip a square piece of paper with dimensions of 120 mm × 120 mm and a weight of 1.2 g is shown in Figure 9. This exhibits the design’s adaptability and flexibility by demonstrating that flat paper can be effectively captured in both an inflated and deflated state.

#### 3.2.2. Case Study: A Three-Fingered VSEAF Gripper

In Figure 10a, a three-fingered VSEAF gripper is shown being designed. For the case study, the three VSEAFs are evenly spaced out on a circular plate. The total weight of the gripper on the VSEAF was 340.2 g. The gripper uses both fingertip and envelope clutching as various grabbing methods. It is referred to as being gripped using the fingertip technique when something is held utilising the tip of the gripper. The grip is mostly produced by friction since there is very little contact area between the tip and the surface of the item. A holder that takes items by the whole package is referred to as an “envelope grab”. This kind of holder has a greater contact area and depends mostly on bending the holder to provide contact friction. As a consequence, Figure 10 shows how the envelope grab may result in larger attempts to grasp and grab for volume and quality of huge things.

As illustrated in Figure 11(a1,a2), a grasping force test rig was set up, the three-fingered VSEAF gripper was mounted vertically, a digital force gauge was used to measure the gripping force, and two balls with a diameter of 20 mm and 40 mm were utilised as substrates Figure 11(b1,b2). This was done to further evaluate the grasping capacity of the three-fingered VSEAFs gripper using these two methodologies. The digital force gauge was used to pull out from the VSEAFs following inflation to obtain the maximum grasping force. Within the VSEAFs, the pressure varied from 20 kPa to 60 kPa in 10 kPa increments.

Five tests were conducted for the VSEAF soft state and stiff state under various inflation pressures. Figure 11(a2,b2) shows the maximum gripping force test results. The rigid state forces of the VSEAFs were larger than the soft state forces for both grabbing styles. For both grasping techniques, the rigid state forces of the VSEAFs were greater than the soft state forces. As can be observed, the friction force of fingertip clutching is lower than that of envelope grabbing, resulting in a significantly lower gripping force for fingertip grasping. The three-fingered VSEAF gripper’s grabbing force for an envelope in a rigid state at 60 kPa is 10.74 N, which is greater than the 8.29 N of a soft state. The grabbing force required by the three-fingered VSEAF gripper to seize at a rigid state at 60 kPa is 6.78 N, which is greater than the 4.76 N required at a soft state. The primary reason for this discrepancy is that fingertip grabbing has a substantially smaller contact area than envelope grasping. During the grabbing force test, the temperature was 27.7 °C ± 0.1 °C, the relative humidity was 60% ± 1%, and the ambient pressure was 1025.5 hPa ± 0.2 hPa, as measured by a meteorological station.

Figure 12 depicts the results of our ongoing research on the ability to grab objects of varying sizes, weights, and shapes (a–d). This indicates that the VSEAF’s gripper can effectively grasp hard and delicate objects, such as apples and oranges (or such as balloons and eggs).

## 4. Conclusions and Future Work

We proposed a pneumatic gripper by integrating a variable stiffness EA pad. The proposed gripper combines the benefits of electroadhesion and soft pneumatic grippers, as well as the capability of changing stiffness. This gripper can grab both flat and convex objects, and its stiffness may be adjusted to boost the gripping stability.

This study presents a simple and lightweight pneumatic electro-adhesive gripper with variable stiffness. The primary contributions of this study are (1) the creation of the idea of a variable stiffness pneumatic electroadhesive gripper, (2) the variable stiffness EA gripper manufacturing process, and (3) the development of a three-fingered VSEAF gripper.

Due to the temperature regulation of the VSEAF, the VSEAF in this research cannot grab high-temperature objects (above 47 °C), necessitating the need for an insulator in future designs. Additionally, future research will involve modeling and optimizing the VSEAF’s structure for enhanced performance, such as a greater EA force, using modeling and simulation.

## Figures and Tables

**Figure 1 polymers-14-04469-f001:**
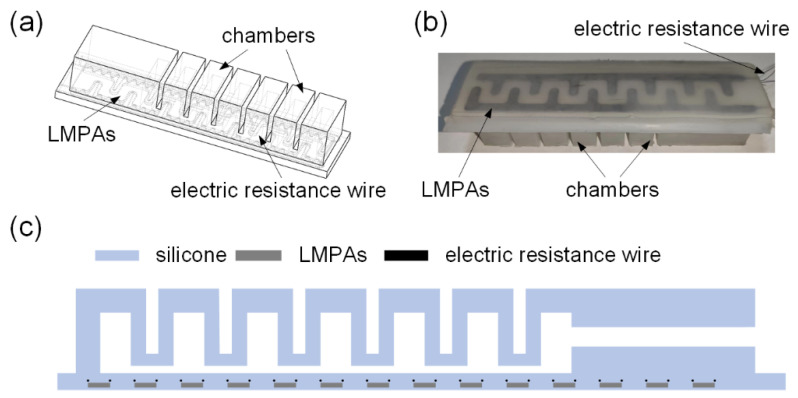
Schematic diagram and the prototype of the VSEAF: (**a**) schematic diagram of the VSEAF in 3D, (**b**) the prototype of VSEAF, and (**c**) schematic diagram of the VSEAF in 2D.

**Figure 2 polymers-14-04469-f002:**
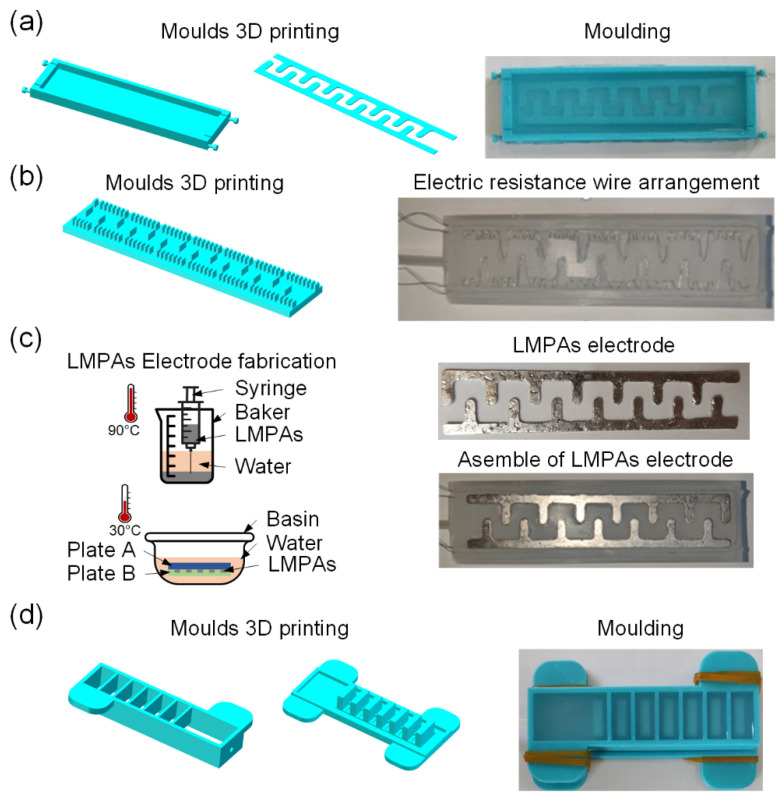
The fabrication procedure of the VSEAF: (**a**) comb groove modeling, (**b**) arrangement of the electric resistance wire, (**c**) fabrication of comb structure low-melting-point alloys, and (**d**) fabrication of a soft pneumatic gripper.

**Figure 3 polymers-14-04469-f003:**
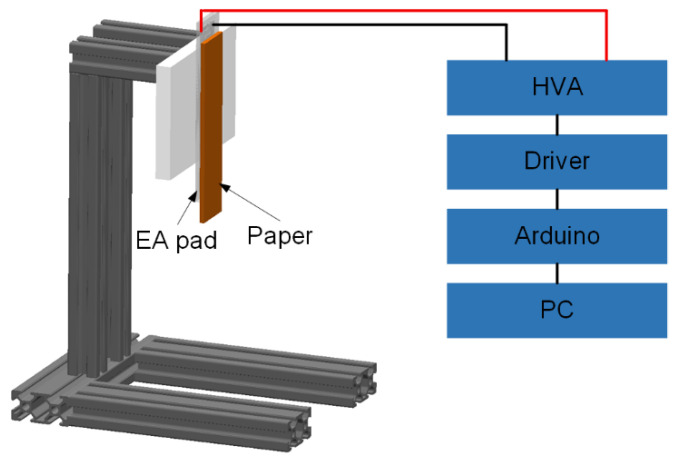
Schematic diagram of the tangential force measurement device.

**Figure 4 polymers-14-04469-f004:**
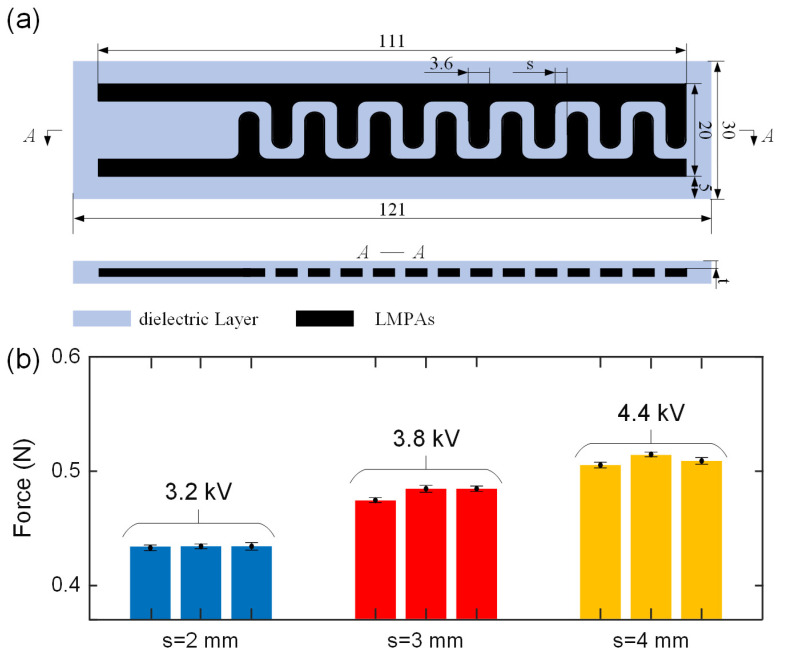
Tangential EA force repeatability test: (**a**) the EA pad design and dimension, and (**b**) measured tangential forces of nine EA pads with three space geometry. Means and one standard deviation for three tests are shown.

**Figure 5 polymers-14-04469-f005:**
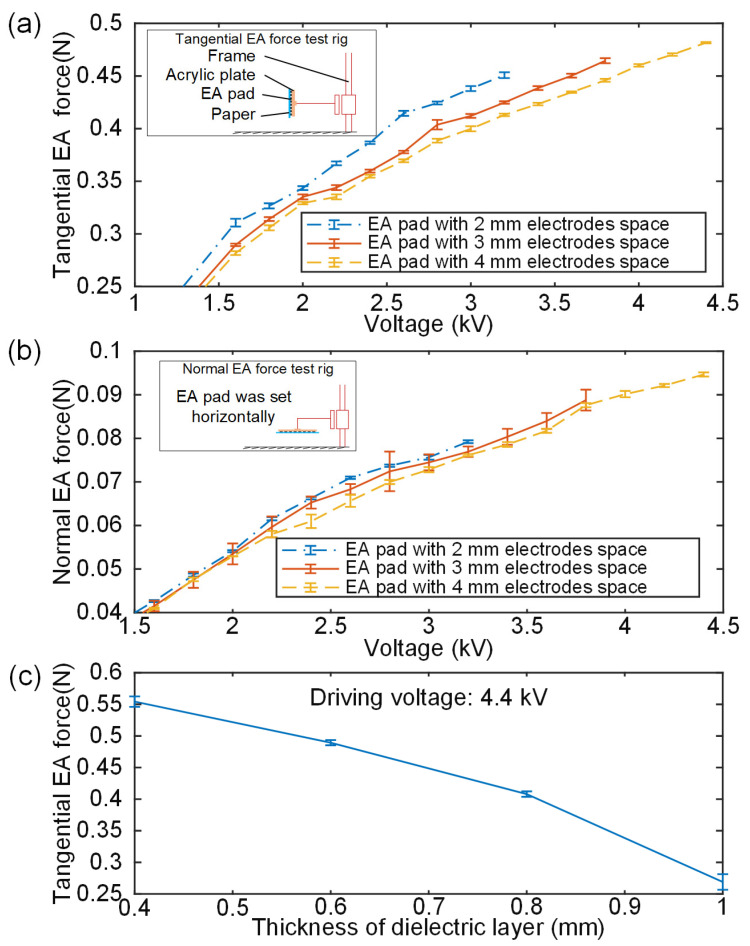
EA pad geometric parameter selection test: (**a**) tangential force test results. Inset shows the schematic diagram of the tangential EA force tests and (**b**) normal EA force test results. Inset shows the schematic diagram of the normal EA force tests and (**c**) tests of tangential EA force on four EA pads with varying dielectric layer thicknesses.

**Figure 6 polymers-14-04469-f006:**
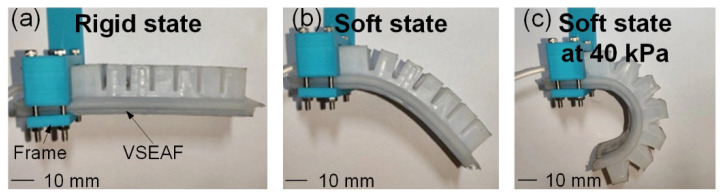
The VSEAF’s bent state under rigid and soft states: (**a**) the VSEAF in a rigid state, (**b**) the VSEAF in a soft state, and (**c**) the VSEAF in a soft state under 40 kPa.

**Figure 7 polymers-14-04469-f007:**
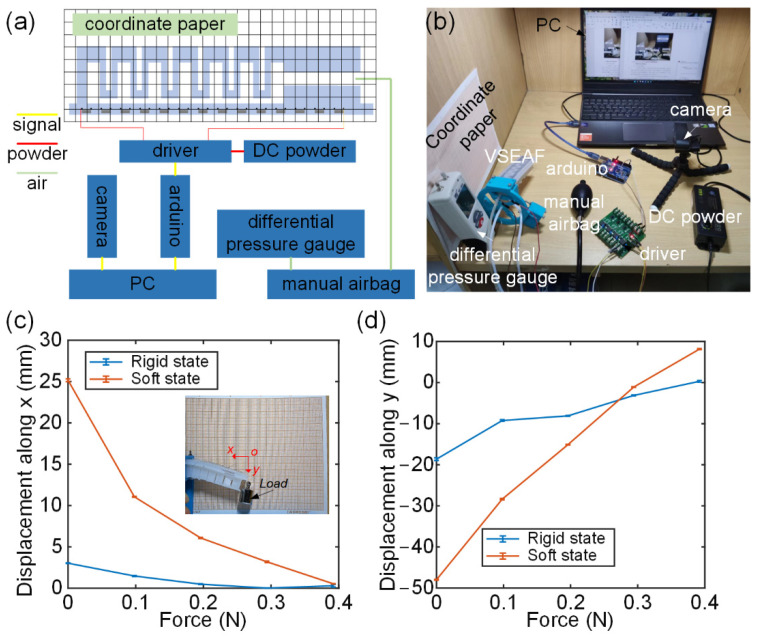
Stiffness performance test rig: (**a**) system diagram of the stiffness performance test, (**b**) the physical setup, (**c**) VSEAF displacement along the *x*-axis under varied loads, (**d**) VSEAF displacement along the *y*-axis under varied loads.

**Figure 8 polymers-14-04469-f008:**
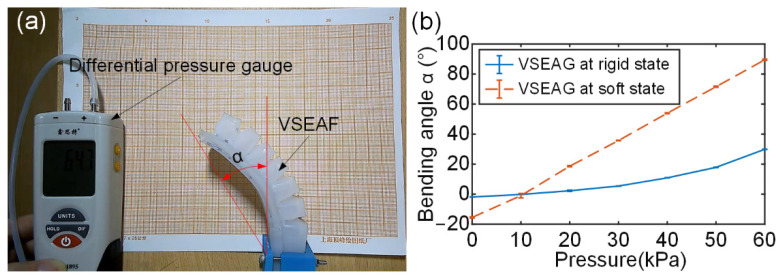
Bending performance test of the VSEAF: (**a**) test rig, (**b**) test results.

**Figure 9 polymers-14-04469-f009:**
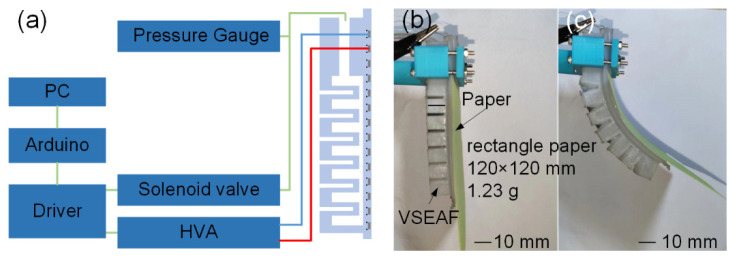
Schematic diagram demonstrating grasping capabilities: (**a**) presentation of the EA gripping control system, (**b**) demonstration of flat paper clutching, and (**c**) demonstration of curved paper grasping.

**Figure 10 polymers-14-04469-f010:**
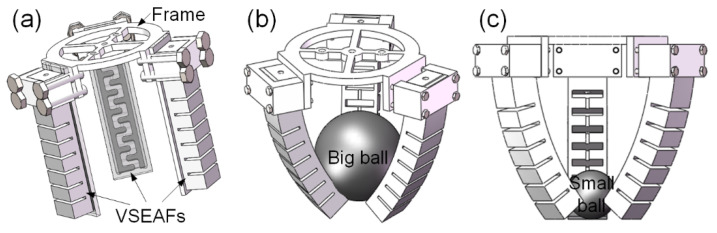
Schematic diagram demonstrating grasping capabilities: (**a**) the designed three-fingered VSEAFs gripper, (**b**) envelope grasping and (**c**) utilizing a fingertip to seize.

**Figure 11 polymers-14-04469-f011:**
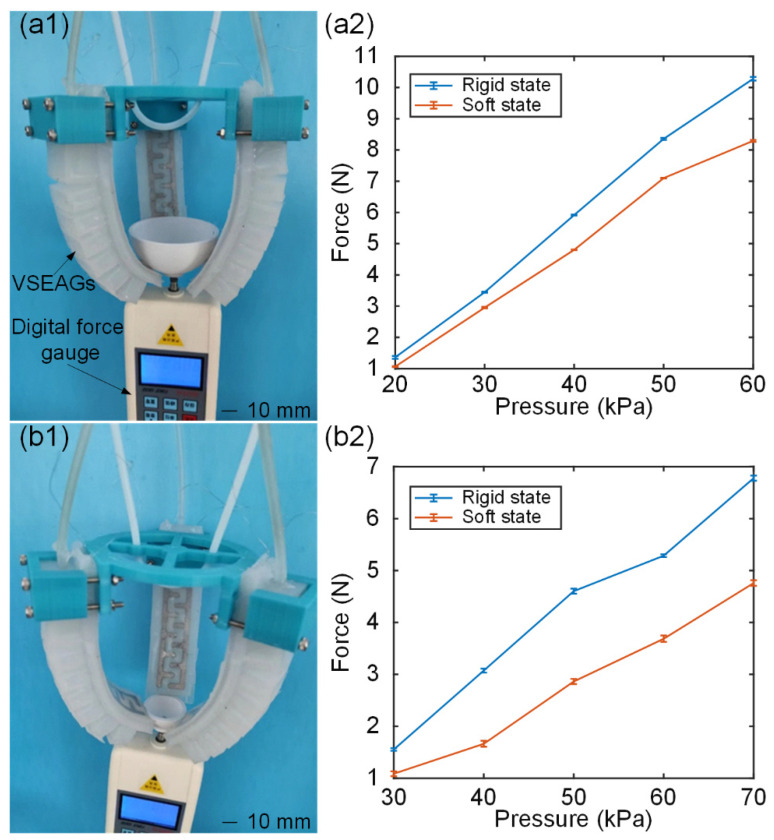
Gripping capability test: (**a1**,**a2**) are the envelope grasping test rig and test results, (**b1**,**b2**) are the fingertip grasping test rig and test results.

**Figure 12 polymers-14-04469-f012:**
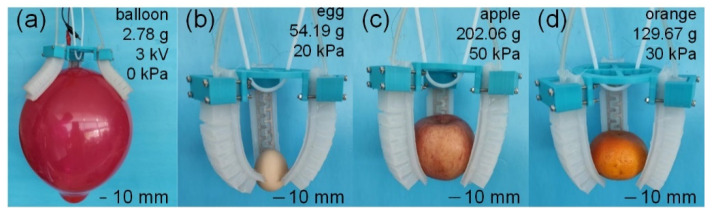
Demonstration of gripping capability. (**a**) shows the EA gripping capability; (**b**–**d**) shows the pneumatic grasping capability.

## Data Availability

Not applicable.

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
