# Peer review of "A Variable Stiffness Electroadhesive Gripper Based on Low Melting Point Alloys"

_polymers, 2022, doi:10.3390/polym14214469_

Round 1
Reviewer 1 Report
Xiang et al have demonstrated fabrication of variable stiffness electro-adhesive gripper (VSEAG) and characterize the forces and design parameters for the same. I would recommend the paper for publication after addressing following comments.
Comments
1. The biggest concern that I have the flowability of LMPA when gripper or finger is soft state which is bent state. The LMPA should get displaced. How repeat is the process in terms of forces with respect to multiple cycles. Authors should elaborate over the advantage of VSEA gripper over only pneumatic ones.
2. Authors should possible limitation and scalability of the proposed VSEAG.
3. The force characterization in figure 4b and 5 is not clear and proper explanation should be added.
4. The English needs to be improved. For example, the line 53-55 on page 2 is not clear. The punctuation is not correct.
5. Short form of variable stiffness electro adhesive gripper should be introduced in line 8 page 1, as authors introduce the full form in material and methods and have used short form in introduction.
6. There are multiple places where captions are missing in figures (Fig 5c, Fig. 7c and d, Fig. 10 a-c.
7. In Fig. 11 a2 is missing in the caption.
8. In Fig 12a, what type of pressure is used should be mentioned as it looks like negative pressure.
9. Bending angle test (Fig.8) should be carried out with multiple samples to improve the readership.
10. It would be helpful to support grasping task (Fig. 12) with supporting movies or videos. This will improve the strength of paper and readership.
11. Authors add the source of LMPA.
12. At couple of places “±” is missing. For example, Page 5 (line 153-154)
Author Response
Response to Reviewer 1
Comment 1.1: The biggest concern that I have the flowability of LMPA when gripper or finger is soft state which is bent state. The LMPA should get displaced. How repeat is the process in terms of forces with respect to multiple cycles. Authors should elaborate over the advantage of VSEA gripper over only pneumatic ones.
Response 1.1: Thank you for the constructive suggestions. For the bending angle test, we've already run several samples, and three tests were run for each inflating pressure. Thus, the flowability of the LMPA is ok for several cycle grasping process. A VSEAF's three-fingered gripper was designed to proof that the VSEA gripper can produce more grasping force than normal pneumatic gripper.
Detailed changes:
In the main text file section 3.2.2 Case study: a three-fingered VSEAFs gripper on Page 10:
The three-fingered VSEAFs gripper's grabbing force for an envelope in a rigid state at 60 kPa is 10.74 N, which is greater than the 8.29 N of a soft state. The grabbing force required by the three-fingered VSEAFs gripper to seize at a rigid state at 60 kPa is 6.78 N, which is greater than the 4.76 N required at a soft state.
Comment 1.2: Authors should possible limitation and scalability of the proposed VSEAG.
Response 1.2: Thank you for pointing this out. We have added possible limitation and scalability in the revised document.
Detailed changes:
In the main text file section 4 Conclusions and future work on Page 12:
Due to the temperature regulation of the VSEAF, the VSEAF in this research can-not grab high temperature objects (above 47 °C), necessitating the need for an insulator in future designs. And future research will involve modeling and optimizing the VSEAF's structure for enhanced performance, such as a greater EA force, using mod-eling and simulation.
Comment 1.3: The force characterization in figure 4b and 5 is not clear and proper explanation should be added.
Response 1.3: Thank you for pointing this out. We have clarified and add clearly explanation of the data in the main text.
Detailed changes:
In the main text file section 2.3 Geometric Parameter Selection and Repeatability test EA Pad Fabrication subsection 2.3.1. Repeatability Test for EA Pad Fabrication on Page 5:
For the fabrication repeatability test, EA pads with three unique spacing were fabricated three times,notably, EA pads with three unique spacing were created three times for the fabrication repeatability test, resulting in the manufacturing of nine EA pads. Three tests were conducted for each EA pad, and these experiments were conducted in a clean, constrained setting. In addition, tests were done at a temperature of 22.6°C ± 0.1 °C, relative humidity of 55% ± 1%, and ambient pressure of 1025.5 hPa ± 0.2 hPa using a weather station. There was a maximum relative difference of 4.5% in the average tangential EA forces collected from the nine EA pads, as shown in figure 3(b). The greatest tangential EA forces of 2 mm, 3 mm, and 4 mm electrode spacing EA pads are 0.43 N, 0.48 N, and 0.512 N, respectively. The electrostatic tangential EA forces difference for three EA Pads with identical specifications is between 0.05N, indicating that the EA PAD manufacturing process is very reliable and can manage error within a suitable range. Using cutting-edge EA pad fabrication methods, such as flexible printed circuit board manufacturing processes, might minimize this discrepancy even more[24].
In the main text file section 2.3 Geometric Parameter Selection and Repeatability test EA Pad Fabrication subsection 2.3.1. Repeatability Test for EA Pad Fabrication on Page 6:
As electrode spacing rises, so does the breakdown voltage of the EA pad; for example, electrode spacings of 2 mm, 3 mm, and 4 mm corresponded to breakdown voltages of 3.2 kV, 3.8 kV, and 4.4 kV, respectively. Clearly, in a certain range, EA pad may produce a larger EA force with a wider electrode spacing, but to achieve the same EA force, an EA pad with a wide electrode spacing requires a higher excitation voltage. Similar to how stronger EA forces need more costly high voltage amplification equipment and higher prices, higher voltage also provides more safety risks, therefore increasing electrode spacing is not the sole approach to enhance adsorption capacity.Maximum tangential force generated by EA pads with 2 mm, 3 mm, and 4 mm electrode spacing is 0.45 N, 0.46 N, and 0.48 N, respectively. 0.07 N, 0.09 N, and 0.10 N are the maximum normal forces produced by EA pads with 2 mm, 3 mm, and 4 mm electrode spacing, respectively.
In the main text file section 2.3 Geometric Parameter Selection and Repeatability test EA Pad Fabrication subsection 2.3.1. Repeatability Test for EA Pad Fabrication on Page 6:
Tangential EA force of EA pads with various dielectric layer thicknesses were investigated since the thickness of the dielectric layer can impact the output of the EA force. Four EA pads with dielectric layer thicknesses of 0.4 mm, 0.6 mm, 0.8 mm and 1 mm are constructed accordingly. The tangential force measurement device was used again, five tangential force tests were conducted for each EA pad. It can be seen from figure 5(c). EA force has obvious downward trend with the increase of the thickness of the dielectric layer, especially when the insulation layer thickness is 1 mm. EA pads with dielectric layer thicknesses of 0.4 mm, 0.6 mm, 0.8 mm and 1 mm are driven at the same voltage of 4.4 kV, and the resultant tangential EA forces for the four kinds of EA pads were 0.55 N, 0.489 N, 0.41 N and 0.27 N, respectively. Compared to EA pads with a 1 mm dielectric layer, EA pads with a 0.4 mm dielectric layer may produce 49% more tangential EA force. Considering the transition of a low melting point alloy from liquid to solid, the heat bilges cold shrink phenomenon exists; therefore, choose a certain amount of security when determining the thickness of the insulating layer, in the event that the insulating layer is compromised due to the low melting point alloy's large expansion volume when it is liquid. Consequently, 0.6mm was chosen for the dielectric layer.
Comment 1.4: The English needs to be improved. For example, the line 53-55 on page 2 is not clear. The punctuation is not correct.
Response 1.4: Thank you very much. This paper has been carefully edited, and the grammar, spelling, and punctuation have been verified and corrected where needed.
Comment 1.5: Short form of variable stiffness electro adhesive gripper should be introduced in line 8 page 1, as authors introduce the full form in material and methods and have used short form in introduction.
Response 1.5: Thank you very much. We have added the abbreviation in the abstract.
Detailed changes:
In the main text Abstract:
‘This paper proposes an electroadhesive gripper (VSEAF) with…’
Comment 1.6: 1) There are multiple places where captions are missing in figures (Fig 5c, Fig. 7c and d, Fig. 10 a-c. 2) In Fig. 11 a2 is missing in the caption. 3) In Fig 12a, what type of pressure is used should be mentioned as it looks like negative pressure. 4) Authors add the source of LMPA. 5) At couple of places “±” is missing. For example, Page 5 (line 153-154).
Response 1.6: We much appreciate your suggestions. We have inserted the missing captions and double-checked all of the articles, and the article has been updated accordingly.
Detailed changes:
In the main text file section 2.3.2 EA Pad Geometric Parameter Selection on Page 6:
Figure 5. EA pad geometric parameter selection test: (a) tangential force test results. Inset shows the schematic diagram of the tangential EA force tests, (b) normal EA force test results. Inset shows the schematic diagram of the normal EA force tests, (c) test of tangential EA force on four EA pads with varying dielectric layer thicknesses.
In the main text file section 3.1 Variable stiffness test on Page 8:
Figure 7. Stiffness performance test rig: (a) system diagram of stiffness performance test, (b) physical setup, (c) VSEAF displacement along the x-axis under varied loads, (d) VSEAF displacement along the y-axis under varied loads.
In the main text file section 3.2.2 Case study: a three-fingered VSEAFs gripper on Page 9:
Figure 9. Demonstration of grasping capabilities schematic diagram: (a) presentation of the EA gripping control system, (b) demonstration of flat paper clutching, and (c) demonstration of curved paper grasping.
In the main text file section 3.2.2 Case study: a three-fingered VSEAFs gripper on Page 9:
Figure 10. Demonstration of grasping capabilities schematic diagram: (a) the designed three-fingered VSEAFs gripper, (b) envelope grasping and (c) utilize fingertip to seize.
In the main text file section 3.2.2 Case study: a three-fingered VSEAFs gripper on Page 10:
Figure 11. Gripping capability test: (a1) and (a2)
In the main text file section 3.2.2 Case study: a three-fingered VSEAFs gripper on Page 10:
Figure 12. Demonstration of gripping capability. (a) demonstrate the EA's grabbing capacity, (b-d) the pneumatic grasping capacity.
In the main text file section 2.1 VSEAF Design on Page 2:
‘…low melting point alloys (LMPAs, indium tin alloy, melting point of 47 °C),…’
In the main text file section 2.3.1. Repeatability Test for EA Pad Fabrication on Page 5:
In addition, tests were done at a temperature of 22.6°C ± 0.1 °C, relative humidity of 55% ± 1%, and ambient pressure of 1025.5 hPa ± 0.2 hPa using a weather station.
In the main text file section 3.2.2. Case study: a three-fingered VSEAFs gripper on Page 10:
Using a meteorological station, the temperature was 27.7 °C ± 0.1 °C and the relative humidity was 60 % ± 1%, and ambient pressure of 1025.5 hPa ± 0.2 hPa during the peak grabbing force.
Comment 1.7: Bending angle test (Fig.8) should be carried out with multiple samples to improve the readership.
Response 1.7: For the bending angle test, we've already run several samples, and three tests were run for each inflating pressure.
Detailed changes:
In the main text file section 3.1. Variable stiffness teston Page 9:
Three tests were run for each inflating pressure of the VSEAF.
Figure 8. Bending performance test of the VSEAF: (a) test rig, (b) test results.
Comment 1.8: It would be helpful to support grasping task (Fig. 12) with supporting movies or videos. This will improve the strength of paper and readership.
Response 1.8: Thank you very much for your constructive suggestions. Due to the lockdown at our university, it is very difficult for us to get the hardware for our equipment. Figure 12 displays the grabbing capabilities of the three-fingered VSEAF gripper with an open-loop control; hence, the pressures were set by trial-and-error experiments, and the pressure for grasping tasks was specified.

Reviewer 2 Report
The study on electroadhesive gripper and Variable of its stiffness is worthfull. However following points to be included for the strengthening of the paper:
1. Modify the title. Do not put any abbreviation in the title for the reader diversity.
2. Use same wording, either "electro-adhesive " or "electroadhesive " throughout the manuscript.
3. Lack of readability as in the 1st sentence of introduction.
4. Page 2, line 65-71: Here there should be objective of the paper. No need to discuss on the construction of the objective in the paper.
5. What is VSEAG design? as present in the last paragraph of the introduction.
Author Response
Response to Reviewer 2
Comment 2.1: 1) Modify the title. Do not put any abbreviation in the title for the reader diversity. 2) Use same wording, either "electro-adhesive " or "electroadhesive " throughout the manuscript. 3) Lack of readability as in the 1st sentence of introduction.
Response 2.1: Thank you very much for pointing this out. We have corrected the title, make the name consistent and rewrite the 1st sentence.
Detailed changes:
In the main text file section Title on Page 1:
Variable stiffness electroadhesive gripper based on low melting point alloys
In the main text file section 1 Introduction on Page 1:
To operate safely in unstructured environments and in collaboration with humans, a new generation of robots must be created; therefore, soft robots comprised of soft materials invented recently have a high degree of deformation, and their interactions with humans and objects are more flexible and compliant than those of traditional stiff robots [1]-[3].
Comment 2.2: Page 2, line 65-71: Here there should be objective of the paper. No need to discuss on the construction of the objective in the paper.
Response 2.2: Your construction recommendations are much appreciated. We have added the objective in the paper.
Detailed changes:
In the main text file section Introduction on Page 2:
To address the soft robots' lack of rigidity and limited load capacity, this paper proposes an electroadhesive gripper (VSEAF) with variable stiffness and a simple construction based on low melting point alloys (LMPAs) with active form adaptation via pneumatic drive.
Comment 2.3: What is VSEAG design? as present in the last paragraph of the introduction.
Response 2.3: Thank you very much for your comments. The VSEAG design is an electroadhesive gripper design, and this VSEAG possesses variable stiffness functionality and can conform to different shapes under pneumatic actuation. Figure 12 demonstrates the three-fingered VSEAF gripper's grabbing capability with an open-loop control, thus the pressures were predetermined through trial and error tests.

Reviewer 3 Report
Dear Authors,
in your interesting manuscript, the following points should be added/changed to further improve it:
- Headline: What does LMPA mean?
- line 53-54: "melting‐point materials [22] is alloy Phase‐change material is a new type of variable stiffness strategy" - this sentence was surely meant differently.
- "The reminder of this article is arranged as follows." - here "reminder" is surely not the right word.
- "2.1. VSEAF Design" - please define abbreviations where they are used for the first time, not in the next sentence.
- Fig. 1c: silicone, not silicon.
- line 85: What does heating have to do with an electro-adhesive force?
- line 96: Why is the alloy not heated by this high voltage?
- Fig. 2b: Where exactly are the resistance wires? And what is the grey material? Apparently not the mold, at least this should be blue according to Fig. 2a. Besides, what does the sketch on the left side show which is not identical to the right side?
- Fig. 2c: I assume you mean a beaker, not a baker. What is shown in "assembly", what is different from "LMPA(s?) electrode"?
FIg. 2: Where in the molds are the previously prepared structures?
- Fig. 3 and text above: What is HVA?
- lines 153 and 154: The "plus minus" signs are missing.
- Fig. 4b: What is S? And where can we see the different geometries? Fig. 4a shows only one of them. Besides, the height is not given in Fig. 4a.
- line 164: Here you mention "optimization", while according to the previous sub-section, you tested just three (apparently randomly chosen) geometries.
- Fig. 5: What is the electrode space, and what the dielectric layer thickness? Besides, why are results shown in Section 2?
- line 200 ff: "Design engineers created four EA pads ..." is doubled.
- Fig. 6: Where are the 40 kPa applied? The text below belongs into Section 2.
- Fig. 8b: The y-axis is not equidistantly spaced. How do the mentioned angles correspond to the image in Fig. 8a? And why does the rigid state not reach an angle of 0°?
- Fig. 9b: What should be visible here?
- Fig. 10: Here the sub-images are not labelled, either.
- According to Fig. 11, the difference between fingertip and envelope grasping is just the geometry of the object to be grasped. This needs more information. Besides, what is different between your systems and any other pneumatic gripper?
- line 320: "This paper ..." is surely "This gripper ...".
- Fig. 12: How were the pressures chosen?
Author Response
Response to Reviewer 3
Comment 3.1: Headline: What does LMPA mean?.
Response 3.1: Thank you for pointing this out. We have added the full name of LMPA in the title according to Response 2.1. LMPA is low melting point alloys.
Comment 3.2: line 53-54: "melting‐point materials [22] is alloy Phase‐change material is a new type of variable stiffness strategy" - this sentence was surely meant differently.
Response 3.2: Thank you for the constructive suggestions. We have corrected this sentence to be consistent. We also checked the whole paper and corrected the errors.
Detailed changes:
In the main text file section 1 Introduction on Page 2:
‘…Using phase-change material is a novel variable stiffness technique, and robots based on this material can alter their stiffness by joule heating directly, resulting in simple construction and straightforward application [22]. In this paper, low-melting-point alloys with good conductivity and variable stiffness characteristics [22] are used as the variable stiffness materials…‘
Comment 3.3: 1) "2.1. VSEAF Design" - please define abbreviations where they are used for the first time, not in the next sentence. 2) Fig. 1c: silicone, not silicon. 3) "The reminder of this article is arranged as follows." - here "reminder" is surely not the right word.
Response 3.3: Thanks. We have explained the acronyms in the introduction and rectified the misspelled terms throughout the document.
Detailed changes:
In the main text file section 1 Introduction on Page 2:
‘…this paper proposes an electroadhesive gripper with variable stiffness and a simple construction (VSEAF) based on low melting point alloys (LMPAs) with active form adaptation via pneumatic drive.’
In the main text file section 2.1 VSEAF Design on Page 2:
In the main text file section 1 Introduction on Page 2:
‘…The contents are organized as follows: …’
Comment 3.4: 1) line 85: What does heating have to do with an electro-adhesive force? 2) Why is the alloy not heated by this high voltage?
Response 3.4: Thank you for your comments. We did not descript this clearly, and the EA working theory is that applying a high voltage (between 0.2 and 20 kilovolts) between electrodes inserted in a dielectric substance creates electrostatic adhesion (EA). This strong electric field creates electrically controlled adhesion on conducting, semiconductive, and insulating surfaces. Electric resistance wires were used as heater through low voltage, and low voltage was enough to directly drive the resistance wire.
Detailed changes:
In the main text file section 1 Introduction on Page 2:
Electrostatic adhesion (EA) is generated by applying a high voltage (typically between 0.2 and 20 kilovolts) across pairs of electrodes embedded in a dielectric material. This high electric field produces an electrically controllable adhesion force on conductive, semi-conductive, and insulating substrates [23].
Comment 3.5: 1) Fig. 2b: Where exactly are the resistance wires? And what is the grey material? Apparently not the mold, at least this should be blue according to Fig. 2a. Besides, what does the sketch on the left side show which is not identical to the right side? 2) Fig. 2c: I assume you mean a beaker, not a baker. What is shown in "assembly", what is different from "LMPA(s?) electrode"? 3) FIg. 2: Where in the molds are the previously prepared structures? 4) Fig. 3 and text above: What is HVA?
Response 3.5: Thanks for your comments. Figures 2 and 3 were not well explained, therefore we changed them. In figure 3, HVA refers to the high voltage amplifier, and its entire name has been included for clarity.
Detailed changes:
In the main text file section 2.2. VSEAF Fabrication on Page 4:
Figure 2. The fabrication procedure of the VSEAF: (a) comb groove modeling, (b) arrangement of the electric resistance wire, (c) fabrication of comb structure low melting point alloys, and (d) fabrication of a soft pneumatic gripper.
In the main text file section 2.3.1. Repeatability Test for EA Pad Fabrication on Page 5:
Figure 3. Schematic diagram of the tangential force measurement device.
Comment 3.6: lines 153 and 154: The "plus minus" signs are missing.
Response 3.6: Thank you for pointing this out. We have inserted the missing captions and double-checked all of the articles, and the article has been updated accordingly according to Response 1.6.
Comment 3.7: 1) Fig. 4b: What is S? And where can we see the different geometries? Fig. 4a shows only one of them. Besides, the height is not given in Fig. 4a. 2) Fig. 5: What is the electrode space, and what the dielectric layer thickness? Besides, why are results shown in Section 2?
Response 3.7: Thanks. We apologize for the confusion; s represents the distance between these two electrodes. EA is one kind of parallel capacitors with a constant distance between electrodes. The entire width of these two electrodes is 20 mm, and we can determine their height. The updated document includes the description. The dielectric layer is on top of the EA pad, which is used for isolation. Figure 4 depicts the dielectric layer thickness, denoted by t. EA pad design relies on the selection of geometric parameters; a well-designed EA pad may create a greater EA force for grabbing of section 3.
Detailed changes:
In the main text file section 2.3.1. Repeatability Test for EA Pad Fabrication on Page 5:
EA is one kind of parallel capacitors, and the space between two electrodes, s, is constant. The spacing, s, between three EA pads is chosen to be 2 mm, 3 mm, and 4 mm, respectively.
Comment 3.8: 1) line 164: Here you mention "optimization", while according to the previous sub-section, you tested just three (apparently randomly chosen) geometries. 2) line 200 ff: "Design engineers created four EA pads ..." is doubled. 3) Fig. 6: Where are the 40 kPa applied? The text below belongs into Section 2.
Response 3.8: Thank you for the constructive suggestions. We have replaced 'optimization' with 'selection'. The phrase 'Design engineers designed four EA pads' has been updated. In the figure 6 (c), 40 kPa air was applied to the VSEAF’ chamber. We amended the captions for figure 6 since they did not adequately convey the picture.
Detailed changes:
In the main text file section 2.3.2. EA Pad Geometric Parameter Selection on Page 7:
‘…the geometric selection of the EA pair unit was based on an experimental investigation employing…’
In the main text file section 2.3.2. EA Pad Geometric Parameter Selection on Page 7:
EA pads with dielectric layer thicknesses of 0.4 mm, 0.6 mm, 0.8 mm and 1 mm are driven at the same voltage of 4.4 kV, and the resultant tangential EA forces for the four kinds of EA pads were 0.55 N, 0.489 N, 0.41 N and 0.27 N, respectively.
In the main text file section 2.3.2. EA Pad Geometric Parameter Selection on Page 8:
Figure 6. The VSEAF's bent state under rigid and soft state: (a) the VSEAF at rigid state, (b) the VSEAF at soft state, and (c) the VSEAF at soft state under 40 kPa.
Comment 3.9: Fig. 8b: The y-axis is not equidistantly spaced. How do the mentioned angles correspond to the image in Fig. 8a? And why does the rigid state not reach an angle of 0°?
Response 3.9: Thank you for your suggestions. We have amended figure 8 and indicated the angle at which the pressure is at its maximum. As a result of gravity, the VSEAF in its soft condition displays bending action.
Detailed changes:
In the main text file section 3.1. Variable stiffness test on Page 10:
The VSEAF in its soft form demonstrates bending movement due to gravity.
In the main text file section 3.1. Variable stiffness test on Page 10:
Figure 8. Bending performance test of the VSEAF: (a) test rig, (b) test results.
Comment 3.10: 1) Fig. 9b: What should be visible here?. 2) Fig. 10: Here the sub-images are not labelled, either. 3) Fig. 12: How were the pressures chosen?
Response 3.10: Thank you for your comments. In figure 9 (b), we want to illustrate the EA's gripping capabilities, demonstrating that the VSEAF can not only grab flat paper, but also curved paper. We apology that figure 10 lacks a label, and have now added them and reviewed the whole paper. Figure 12 demonstrates the three-fingered VSEAF gripper's grabbing capability with an open-loop control, thus the pressures were predetermined through trial and error tests.
Detailed changes:
In the main text file section 3.2.2 Case study: a three-fingered VSEAFs gripper on Page 10:
Figure 10. Demonstration of grasping capabilities schematic diagram: (a) the designed three-fingered VSEAFs gripper, (b) envelope grasping and (c) utilize fingertip to seize.
Comment 3.11: According to Fig. 11, the difference between fingertip and envelope grasping is just the geometry of the object to be grasped. This needs more information. Besides, what is different between your systems and any other pneumatic gripper?
Response 3.11: Thank you for pointing them out. I appreciate your pointing them out. The reviewer is correct; in its soft condition, the three-fingered VSEAFs gripper is not much different from other common pneumatic grippers. However, if the VSEAF's three-fingered gripper in its rigid form can provide more grabbing force. The comparison of the three-fingered VSEAF gripper's gripping force has previously been conducted.
Detailed changes:
In the main text file section 3.2.2 Case study: a three-fingered VSEAFs gripper on Page 10:
The three-fingered VSEAFs gripper's grabbing force for an envelope in a rigid state at 60 kPa is 10.74 N, which is greater than the 8.29 N of a soft state. The grabbing force required by the three-fingered VSEAFs gripper to seize at a rigid state at 60 kPa is 6.78 N, which is greater than the 4.76 N required at a soft state.
Comment 3.12: line 320: "This paper ..." is surely "This gripper ...".
Response 3.12: Thank you very much. We regret for the error and have rewritten and reviewed the whole document.
Detailed changes:
In the main text file section 4. Conclusions on Page 12:
This gripper can grab both flat and convex objects, and its stiffness may be adjusted to boost the gripping stability.

Round 2
Reviewer 1 Report
Authors have addressed all reviewer's comments. I recommend for publication.
Reviewer 3 Report
Dear Authors,
after revising the manuscript, I would now suggest accepting it.